# HIV-Infected Hepatic Stellate Cells or HCV-Infected Hepatocytes Are Unable to Promote Latency Reversal among HIV-Infected Mononuclear Cells

**DOI:** 10.3390/pathogens13020134

**Published:** 2024-02-01

**Authors:** Cinthya Alicia Marcela López, Rosa Nicole Freiberger, Franco Agustín Sviercz, Jorge Quarleri, María Victoria Delpino

**Affiliations:** Instituto de Investigaciones Biomédicas en Retrovirus y Sida (INBIRS), Facultad de Medicina, Consejo de Investigaciones Científicas y Técnicas (CONICET), Universidad de Buenos Aires, Buenos Aires 1121, Argentina; alilopez1996@gmail.com (C.A.M.L.); freibergernicole@gmail.com (R.N.F.);

**Keywords:** HIV, HCV, co-infection, J-Lat, U1, latency reversal, liver

## Abstract

Due to a common mode of transmission through infected human blood, hepatitis C virus (HCV) and human immunodeficiency virus (HIV) co-infection is relatively prevalent. In alignment with this, HCV co-infection is associated with an increased size of the HIV reservoir in highly active antiretroviral therapy (HAART)-treated individuals. Hence, it is crucial to comprehend the physiological mechanisms governing the latency and reactivation of HIV in reservoirs. Consequently, our study delves into the interplay between HCV/HIV co-infection in liver cells and its impact on the modulation of HIV latency. We utilized the latently infected monocytic cell line (U1) and the latently infected T-cell line (J-Lat) and found that mediators produced by the infection of hepatic stellate cells and hepatocytes with HIV and HCV, respectively, were incapable of inducing latency reversal under the studied conditions. This may favor the maintenance of the HIV reservoir size among latently infected mononuclear cells in the liver. Further investigations are essential to elucidate the role of the interaction between liver cells in regulating HIV latency and/or reactivation, providing a physiologically relevant model for comprehending reservoir microenvironments in vivo.

## 1. Introduction

According to World Health Organization (WHO), there are an estimated 39.0 million people living with HIV, and 29.8 million people living with HIV who are receiving antiretroviral therapy [1]. Those infected with HIV and receiving HAART experience an increased life expectancy, marked by a reduced occurrence of AIDS-related morbidity and mortality [2]. Patients on HAART typically survive over 10 years post-AIDS onset, while those without it often succumb within just 2 years [3].

While the treatment has proven highly effective in suppressing viremia [4], addressing the persistence of HIV in latent tissue reservoirs remains a significant hurdle for long-term management [5,6].

HIV primarily targets cells within the lymphoid and myeloid lineages, such as T-helper lymphocytes and monocyte-derived macrophages. Within these cells, the virus genome is retrotranscribed, and then integrated into the host DNA, forming the provirus [7]. HIV persists in reservoirs that are largely resistant to the effects of HAART [8]. Viral reservoirs mainly consist of CD4+ T cells but also macrophages containing transcriptionally silent yet potentially inducible replication-competent proviruses located in multiple anatomical sites wherein replication-competent forms of the virus endure with more stable kinetic properties than in the primary pool of the actively replicating virus. However, these host cells are also transcriptionally programmed to enter a quiescent state, which is conducive to HIV latency. Interestingly, the presence of myeloid cells in co-culture with activated HIV-infected T cells may enhance their transition to a post-activation state of latency, underscoring the role of cell-to-cell contact in the establishment of HIV latency [9].

A better understanding of host factors and physiological signaling pathways governing latency and reactivation within HIV reservoirs in tissues could pave the way for the development of safer and more efficacious therapeutic approaches for individuals living with HIV [10].

Due to a common mode of transmission through infected human blood, hepatitis C virus (HCV) and HIV co-infection is relatively prevalent, with an estimated 2.3 million individuals globally living with HCV/HIV co-infection [11].

While HCV primarily targets the liver, chronic HCV infection also involves significant propagation in extrahepatic sites, with detection in serum and peripheral blood mononuclear cells. Given that CD4+ T cells serve as the primary site for HIV replication, the co-infection of these cells can give rise to intricate interactions between both viruses [12,13]. Numerous studies have shown that HIV infection speeds up the progression of hepatic fibrosis caused by HCV infection. The mechanisms behind the accelerated hepatic fibrosis in individuals co-infected with HIV and HCV remain unclear but are likely multifaceted, involving factors such as direct viral impact, immune/cytokine dysregulation, and heightened oxidative stress [14,15,16].

The sustained presence and renewal of latently infected cells result from various mechanisms, including cellular activation, clonal expansion, and homeostatic processes. These processes may hold specific significance, as they can be modulated within the context of co-infection [17].

Prior studies have shown that inflammation plays a crucial role in the signal-dependent transcription of HIV [18,19]. Increased inflammatory response was noted in patients with HCV/HIV co-infection when compared with those with either HCV or HIV mono-infection [20,21]. In concordance, HCV co-infection is related to an increased HIV reservoir size in HAART-treated HIV individuals [22]. 

The possible mechanisms of interaction between human immunodeficiency virus-infected liver parenchymal and non-parenchymal cells have been studied to understand the development of a fibrotic phenotype [23]. The presence of HIV provirus in the liver was detected in a human autopsy study, revealing several major HIV reservoir cells, such as resting memory CD4+T lymphocytes, dendritic cells, and macrophages [24]. 

Hence, our study investigates the interplay between HCV/HIV co-infection in liver cells and its impact on the modulation of HIV viral latency using latently infected cell lines infected with intact virus (J-Lat and U1).

## 2. Materials and Methods

### 2.1. Cell Culture

The spontaneously immortalized human hepatic stellate cell line (LX-2) was generously provided by Dr. Scott L. Friedman (Mount Sinai School of Medicine, New York, NY, USA). LX-2 cells were maintained in Dulbecco’s Modified Eagle Medium (DMEM, Life Technologies, Grand Island, NY, USA), supplemented with 5% fetal bovine serum (FBS; Life Technologies), L-glutamine (2 mM), 100 U/mL penicillin, and 100 µg/mL streptomycin. To study HSC transdifferentiation, LX-2 cells were cultured in DMEM supplemented with 2% FBS. The J-Lat 10.6 cell line is a subclone derived from Jurkat-based cells infected with a pseudotyped human immunodeficiency virus type 1 (HIV-1) (genus *Lentivirus*, family *Retroviridae*) strain, HIV/R7/E−/GFP [25,26]. Chronically infected HIV-1 promonocytic (U1) cell lines are clones derived through limiting dilution cloning of U937 cells that survived an acute infection with HIV-1 (LAV-1 strain), initially generated by Folks et al. [27]. Both were obtained from the NIH AIDS Reagent Program, Division of AIDS (NIAID, NIH).

THP-1 monocyte cell line and Jurkat cell (immortalized T lymphocytes) were obtained from the American Type Culture Collection (Manassas, VA, USA). THP-1 and J-Lat cells were cultured in RPMI 1640 as previously described. J-Lat and U1 cells were treated with 50 ng/mL phorbol 12-myristate 13-acetate (PMA, Sigma Aldrich, Argentina) as a positive control. Additionally, to demonstrate the ability of an inflammatory stimulus as a latency reversal agent, THP-1 cells were stimulated for 4 h with *Escherichia coli* lipopolysaccharide (LPS), and then pelleted using centrifugation at 180× *g* for 15 min and washed twice in 10 mL of RPMI. Then, these cells were co-cultured with J-Lat cells. Alternatively, culture supernatants from THP-1 cells, stimulated with LPS for 4 h, and then pelleted using centrifugation at 180× *g* for 15 min and washed twice in 10 mL of RPMI and cultured for an additional 18 h, were used.

Cocultures of J-Lat: LX-2, J-Lat: Huh7.5, U1: Huh7.5 cells, and J-Lat: THP1 cells were performed at 1:1 proportion over 24 and 72 h. Stimulation of J-Lat with culture supernatants (conditioned-medium-infected or not) from LX-2, Huh7.5, and THP-1 cells was performed at a 1/2 dilution. Stimulation of U1 with culture supernatants from Huh7.5 conditioned-medium-infected or not was performed at ½ dilution. Latency viral reversion was evaluated at 24 and 72 h post stimulation or cell coculture. 

Azidothymidine (AZT, 20 μM, Sigma-Aldrich, Buenos Aires, Argentina) was used as reverse transcriptase inhibitor.

### 2.2. Viral Stocks

Wild-type (WT) HIV NL43 (X4-tropic) strain was available. We used full-length infectious molecular clones of HIV, pBR-NL4.3 (from Dr. Malcolm Martin), and NLAD8 (from Dr. Eric O. Freed), which were obtained through the NIH AIDS Reagent Program (Division of AIDS, NIAID, NIH, USA) [28,29]. The NL43-VSV-G strain was produced through co-transfection with the proviral plasmid in combination with pVSVG to pseudotype envelope-defective viruses with vesicular stomatitis virus (VSV) glycoprotein G. We co-transfected 293T cells with a VSV-G expression plasmid (pCMV–VSV-G) using an HIV-NL43/VSV-G plasmid ratio of 10:1. After 24 h, we replaced the culture medium and harvested lentiviral particles at 48 and 72 h post-transfection. The supernatants were pre-cleared by centrifugation, and ultra-concentrated for 5 h at 18,000 rpm, and the resulting pellet was resuspended in DMEM supplemented with 10% fetal bovine serum (FBS). We stored the concentrated viral particles at −86 °C until further use. The quantification of HIV capsid (p24 antigen) in the viral stocks was determined using a commercial ELISA assay (INNOTEST^®^ HIV Antigen mAb, Los Angeles, CA, USA). HCV particles were obtained using the J6/JFH clone (from Apath LLC, New York, NY, USA) [30]. The viral stock was amplified via infection of Huh7.5 cells and harvest culture supernatants. Uninfected culture supernatants were used as control. 

HCV RNA load level was determined using a quantitative real-time PCR-based Cobas^®^ HCV Test. HCV RNA was isolated from 400 µL of culture supernatant using the Cobas^®^ 4800 System, which consists of separate devices for sample preparation (Cobas x480) and amplification/detection (Cobas z480 analyzer). The dynamic range of quantification was 15 to 10^8^ IU/mL (1.2–8.0 Log IU/mL). The limit of detection (LoD) was 7.6 IU/mL in serum and 9.2 IU/mL in plasma, and the lower limit of quantification (LLOQ) was 15 IU/mL. 

### 2.3. Cellular Infection

LX-2 cells were challenged with pseudotyped HIV co-expressing the G glycoprotein from vesicular stomatitis virus (VSV-G-HIV). All experiments were conducted in a BSL-3 laboratory at INBIRS. In accordance with institutional rules, all biological materials were mandatorily autoclaved before disposal through incineration. Incineration was carried out in a high-temperature incinerator.

LX-2 cells were seeded at a density of 50,000 cells per well in 24-well plates and exposed to an HIV inoculum of 0.5 pg of p24 per cell. 

Assessment of infectivity and replication involved the measurement of intracellular p24 expression utilizing the KC57 monoclonal antibody labeled with phycoerythrin against p24 (PE-KC57 [FH190-1-1]) protein (Beckman Coulter, Brea, CA, USA) through flow cytometry at 24, 48, and 72 h.

We conducted experiments to determine whether pseudotyped-HIV-infected LX-2 cells can release infectious viral particles. Culture supernatants from VSV-G-HIV-GFP-infected LX-2 cells were harvested at 3 days post-infection. This conditioned media were used to expose permissive Jurkat T cells. Infectivity and replication was evaluated at 3, 5, 7 and 10 days post infection in the presence or absence of azidothymidine (AZT).

Huh7.5 cells were seeded at a density of 50,000 cells per well in 24-well plates and exposed to HCV at a multiplicity of infection of 1. After 4 h of virus exposure at 37 °C, the cells were washed four times with phosphate-buffered saline (PBS) to remove unabsorbed virus and then incubated in fresh culture medium at 37 °C. 

Infectivity and replication was evaluated at 24, 48, and 72 h in culture supernatants from infected-HuH7.5 cells by RT-qPCR. 

RNA extraction was performed using the Chemagic™ Viral DNA/RNA kit special H96 (PerkinElmer, Rodgau, Germany) on the automated Chemagic™ 360 instrument (PerkinElmer, Germany). Quantification of RNA was carried out using a NanoDrop™ (Thermo Scientific™, Waltham, MA, USA). cDNA synthesis was accomplished using the reverse transcriptase enzyme Improm-II (Promega, Wisconsin, WI, USA). Real-time PCR was conducted using the following primers: Forward—TTCACGCAGAAAGCGTCTAG, Reverse—CACTCTCGAGCACCCTATCAGGCAGT. The real-time PCR assay utilized SYBR green as a DNA-binding fluorescent dye on a StepOne Real-Time PCR System (Applied Biosystems, Waltham, MA, USA).

The real-time PCR protocol involved an initial step of 15 min at 50 °C for cDNA synthesis, followed by 5 min at 94 °C for initial denaturation. The cycling phase included 35 cycles with denaturation at 94 °C for 45 s, annealing at 55 °C for 45 s, and extension at 72 °C for 45 s in each cycle.

To validate the results, viral copies were quantified in a culture supernatant quantitative real-time PCR-based Cobas^®^ HCV Test, as described above. Culture supernatants from infected and uninfected cells were harvested at 24, 48, and 72 h post-infection and stored at −70 °C until use.

### 2.4. Determination of HIV Latency Reversal

Latency reversal was determined through flow cytometry, measuring the percentage of cells positive for enhanced green fluorescent protein (eGFP) in J-Lat 10.6 cells. The cells from the chronically infected monocytic U937 cell line (U1) were fixed and permeabilized using the Fixation/Permeabilization Kit (BD Biosciences, Franklin Lakes, NJ, USA) according to the manufacturer’s instructions for 30 min at 4 °C. Low adherence was detected in U1 cells treated with PMA, and they were harvested using a cooled 4 °C physiological solution. Latency reversion was assessed by quantifying intracellular p24 expression, utilizing the KC57 monoclonal antibody labeled with phycoerythrin against p24, known as PE-KC57 (Beckman Coulter, USA) for 1 h at 4 °C. Data were acquired using a FACSCanto II (Becton Dickinson, Franklin Lakes, NJ, USA) and analyzed with FlowJo v10.6.2 (Ashland, Wilmington, DE, USA).

### 2.5. Statistical Analysis

Statistical analysis was performed with one-way ANOVA. Multiple comparisons between all pairs of groups were made with Tukey’s post hoc test, and those against two groups were conducted with Student’s *t*-test and Mann–Whitney test. To determine normality, the Shapiro–Wilk normality test was used. The groups under comparison included latently HIV-infected cells treated with culture supernatants from infected cells versus those treated with culture supernatants from non-infected cells. Additionally, there was a comparison involving the co-culture of latently HIV-infected cells with infected cells versus non-infected cells. Positive controls were compared with non-treated cells. Graphical and statistical analyses were performed with GraphPad Prism version 8.0.1 for Windows, GraphPad Software, Boston, MA, USA Each experiment was performed in triplicate with different culture preparations on five independent occasions. Data were represented as mean ± SD measured in triplicate from three individual experiments. A *p* < 0.05 is represented as *, *p* < 0.01 as **, *p* < 0.001 as ***, and *p* < 0.0001 as ****. A statistically significant difference between groups was accepted at a minimum level of *p* < 0.05.

## 3. Results

### 3.1. HIV-Infected Hepatic Stellate Cells (LX-2) Were Not Able to Reverse Viral Latency in J-Lat Cells

As previously mentioned, HIV genomic RNA levels increase in the serum of patients with liver involvement. HIV infection results in the modulation of cytokines, which play a role in regulating the homeostasis of the immune system [31,32,33]. 

To investigate the modulation of viral latency reversal, culture supernatants from VSV-G-HIV-infected LX-2 cells obtained at 24 and 72 h post-infection were used to stimulate a latently infected T-cell line J-Lat over 24 and 72 h. Our results showed that conditioned media from HIV-infected LX-2 cells were unable to reverse viral latency in J-Lat cells (Figure 1A,B).

Additionally, co-culturing HIV-infected LX-2 cells with J-Lat cells over 24 and 72 h was also unable to reverse viral latency (Figure 1C,D)). However, J-Lat cells were capable of reversing latency when stimulated with PMA, which served as a positive control (Figure 1).

The absence of latency reversion could not be attributable to the absence of viral replication, since VSV-G-HIV vas able to replicate in LX-2 cells (Figure 2A,B). Additionally, experiments were performed to investigate the potential release of infectious viral particles from pseudotyped-HIV-infected LX-2 cells. The culture supernatants obtained from VSV-G-HIV-GFP-infected LX-2 cells were collected at 3 days post-infection. Subsequently, these conditioned media were utilized to expose permissive Jurkat T cells. As illustrated in Figure 2C,D, viral replication in Jurkat cells exhibited a gradual increase from day 3 to day 10 following exposure to supernatants from HIV-pseudotyped-infected LX-2 cells. The pretreatment of Jurkat cells with AZT significantly reduced viral replication, providing additional evidence for the infectivity of the viral progeny released from infected LX-2 cells. 

Furthermore, viral latency was reversed when J-Lat cells were co-cultured with LPS-treated monocytes (THP-1 cells), or exposed to culture supernatants from THP-1 cells that were previously stimulated with LPS (Figure 3). These findings suggest that a similar approach, but with the induction of appropriate stimuli, could potentially reverse viral latency in J-Lat cells. 

Taken together our results indicated that that neither the soluble mediators released by HIV-infected HSCs nor cell to cell contact are capable of promoting latency reversal. 

### 3.2. HCV-Infected Hepatocytes (Huh7.5 Cells) Were Not Able to Reverse Viral Latency in J-Lat Cells and U1 Cells

Huh7.5 cells were infected with HCV and viral infectivity and replication was evaluated in culture supernatants at 24, 48, and 72 h post-infection (Figure 4A).

To determine the role of HCV in modulating latency reversion, Huh7.5 cells were infected with HCV, and culture supernatants were collected 24 and 72 h post-infection. J-Lat cells were then stimulated over 24 and 72 h with culture supernatants from HCV-infected Huh7.5 cells to investigate the possibility of latency reversion. Culture supernatants from uninfected Huh7.5 cells served as a control. Our results indicated that culture supernatants from HCV-infected Huh7.5 cells were unable to induce latency reversion in J-Lat cells (Figure 4B,C). However, J-Lat cells were capable of reversing latency when stimulated with PMA, serving as a positive control (Figure 4). Additionally, co-culturing HCV-infected Huh7.5 cells with J-Lat cells also failed to reverse viral latency (Figure 4D,E).

Besides CD4+ T lymphocytes, cells of the myeloid lineage, especially macrophages, are believed to be important for HIV-1 persistence [34].

An experiment was conducted to determine whether HCV-infected Huh7.5 cells could induce latency reversion in a latently infected monocytic cell, U1. Our results indicated that culture supernatants from HCV-infected Huh7.5 obtained at 24 and 72 h post-infection were unable to induce latency reversion in U1 cells stimulated during 24 or 72 h. Additionally, supernatants from uninfected Huh7.5 cells or the co-culture of uninfected cells with U1 had no effect (Figure 5A,B). Alternatively, co-culture between HCV-infected Huh7.5 and U1 also failed to induce latency reversion at 24 and 72 h post coculture (Figure 5C,D). However, latency reversion was observed when U1 cells were stimulated with PMA (Figure 5). Additionally, supernatants from uninfected Huh7.5 cells or the co-culture of uninfected cells with U1 had no effect.

Collectively, our findings indicate that neither the soluble mediators released by HCV-infected Huh7.5 cells nor the ligands expressed on the membrane of these cells were able to promote latency reversal.

## 4. Discussion

Owing to a common mode of transmission through infected human blood, the co-infection of hepatitis C virus (HCV) and human immunodeficiency virus (HIV) is relatively prevalent, affecting an estimated 2.3 million people worldwide [11].

Both HIV and HCV impact hepatocytes and hepatic stellate cells (HSCs), stimulating the generation of reactive oxygen species (ROS). Consequently, this induces the activation of p38 mitogen-activated protein kinase (MAPK), c-Jun N-terminal kinase (JNK), and extracellular signal-regulated kinases (ERKs), leading to the activation of nuclear factor kappa (NF-κB). These events support the expression of pro-fibrogenic TGF-β1 genes, responsible for encoding collagen and TIMP-1, while concurrently down-regulating the synthesis of MMP-3. Thus, both HCV and HIV play direct roles in liver damage by initiating apoptosis and suppressing the production of antioxidant protective mediators [35,36].

The quantitative and qualitative deterioration of T-cell responses linked to HIV infection can adversely affect the progression of HCV-related diseases. Given the crucial role of the adaptive immune system in clearing HCV and the deleterious impact of HIV infection on T cells, it is unsurprising that HCV persistence is more prevalent in individuals with HIV/HCV co-infection compared with those solely infected with HCV [37].

HCV infection further triggers the activation of macrophages, particularly Kupffer cells, leading to the release of reactive oxygen species (ROS) and substantial amounts of proinflammatory and fibrogenic mediators [38,39] including TGF-β1. Numerous studies have indicated an elevated secretion of TGF-β1 from HCV-infected cells, potentially fueling the activation of hepatic stellate cells (HSCs) and the subsequent progression of hepatic fibrogenesis [40,41]. Additionally, both Kupffer cells and activated human HSCs express TLR4, the primary receptor for lipopolysaccharide (LPS), which is abundantly released during microbial translocation associated with both HCV and HIV infections [42].

In a recent study, a larger size of the HIV reservoir in resting CD4+ T cells was observed among individuals with HCV/HIV co-infection who were undergoing ART treatment. This trend was evident in both individuals with chronic HCV and those who had spontaneously resolved HCV, as compared with subjects infected with HIV alone [22]. Likewise, studies have reported that co-infection with HCV influences the progression of HIV disease in people living with HIV (PLWH) who are undergoing antiretroviral therapies (HAART). In cases of HCV co-infection, there is a detrimental impact on the homeostasis of CD4+ T-cell counts, facilitating HIV replication and contributing to the persistence of viral reservoirs [43].

Numerous studies have consistently demonstrated the pivotal role of latently infected cells in the persistence, propagation, and dissemination of HIV [18,19,23]. Despite the effectiveness of systemic HAART in reducing plasma viral load, it primarily focuses on this aspect and does not specifically target latently infected cells within anatomical reservoirs [44]. The factors that intricately regulate the latency and/or reactivation of HIV within microenvironments of reservoirs remain poorly understood. Liver cells are well-established for their interactions with both macrophages and T cells, influencing their activation phenotype [45,46]. It has been previously demonstrated that hepatic stellate cells and hepatocytes secrete soluble mediators in response to HIV and HCV infection, respectively [47,48,49]. The balance between the pro- and anti-inflammatory mediators create a microenvironment that could be involved in the reactivation or maintenance of HIV latency.

Additionally, latency reversion could be modulated by cell to cell contact in a way dependent on the cell involved and the state of activation. It has been demonstrated that interactions between monocytes/dendritic cells and latently HIV-infected T cells play a crucial role in reversing latency. Conversely, a post-activation T-cell latency model, when in contact with monocytes and subjected to anti-CD3 stimulation, demonstrated a reduction in virus expression [50].

Hence, undertaking further studies to define the specific roles of soluble mediators and cell–cell contact receptors in the maintenance and reactivation of latency could potentially result in a significant breakthrough in understanding the mechanisms that contribute to the modulation of the viral reservoir.

Our findings, employing the latently infected monocytic cell line (U1) and the latently infected T-cell line (J-Lat) revealed that the cytokines produced by the infection of hepatic stellate cells and hepatocytes with HIV and HCV, respectively, were unable to induce latency reversal under the conditions studied, indicating that the microenvironment induced by direct viral interaction with hepatic cells are not responsible for latency reversal.

The liver serves as a secondary lymphoid organ, hosting a significant population of CD4+ T cells and boasting the largest concentration of tissue-resident macrophages in the body. Consequently, the liver might act as a reservoir for HIV, as both HIV DNA and RNA have been detected in human hepatocytes and liver macrophages, persisting even when suppressive HAART is administered. Yet, it is conceivable that the liver microenvironment, particularly in hepatocytes, as opposed to CD4+ T cells with diverse activation statuses, may be favorable to latency [51,52,53].

Additionally, a point to consider is that in the conditioned medium from the HIV-infected hepatic stellate cells, there was a release of an HIV wild type, which is lymphotropic and, although it should have replicative capacity, fails to become a “stimulus” for latently infected J-Lat cells.

Monocytes, which harbor replication-competent viruses, have the potential to replenish tissue macrophage reservoirs upon leaving the bloodstream and undergoing differentiation into monocyte-derived macrophages [54]. Furthermore, due to their ability to transmigrate into tissue compartments, macrophages are implicated in viral dissemination to multiple anatomical sites [55,56]. However, our experiments indicated that HCV-infected hepatocytes were unable to reverse the latency in U1 cells.

Numerous studies have emphasized the crucial role of latently infected cells in HIV-1 persistence, propagation, and dissemination [57,58,59,60]. Despite the primary focus of systemic Highly Active Antiretroviral Therapy (HAART) on reducing plasma viral load, it does not specifically target latently infected cells residing in anatomical reservoirs [61,62,63,64]. The factors that critically regulate the latency and/or reactivation of HIV-1 within reservoir microenvironments remain poorly understood.

To gain insights, investigations into the involvement of tissue-resident hepatic cells in regulating HIV-1 latency and/or reactivation could provide a physiologically relevant model for understanding reservoir microenvironments in vivo. Furthermore, these explorations may contribute to the development of more effective strategies for eliminating persistent HIV reservoirs in patients. Subsequent studies will delve into the roles of resident and infiltrating immune cells, as well as coexposure with latency reversal agents (LARs), aiming to facilitate the discovery of more effective strategies to eliminate persistent HIV-1 reservoirs in patients.

Further investigations are needed to elucidate the role of the interaction between liver cells in regulating HIV latency and/or reactivation, with the goal of providing a physiologically relevant model for understanding reservoir microenvironments in vivo.

## Figures and Tables

**Figure 1 pathogens-13-00134-f001:**
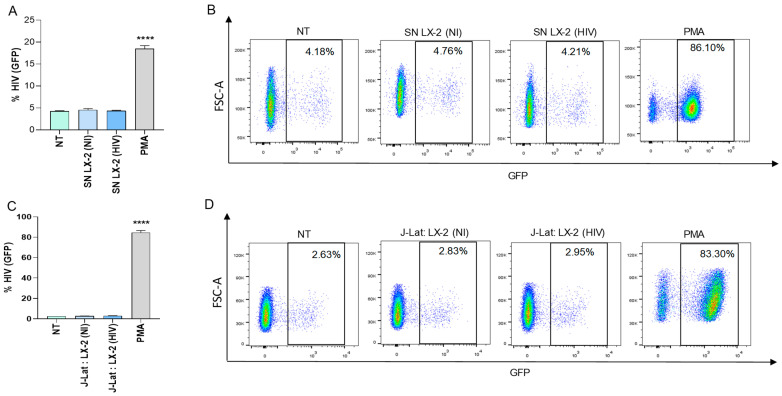
VSV-G-HIV-GFP-infected LX-2 were not able to reverse viral latency in J-Lat cells. J-Lat cells (clone 10.6) were stimulated with culture supernatants from VSV-G-HIV-infected LX-2 cells (0.5 pg of p24 per cell), harvested at 24 h post-infection at 1/2 proportion. Cells stimulated with 50 ng/mL of phorbol 12-myristate 13-acetate (PMA) was used as positive control. Culture supernatants from non-infected LX-2 and THP-1 cells were used as control. At 72 h, latency reversion was quantified as a percentage of GFP positive J-Lat cells (**A**). Representative dot plots obtained by flow cytometry represented in A (**B**). VSV-G-HIV-GFP-infected LX-2 were co-cultured with J-Lat cells at 1:1 proportion. Latency reversion was quantified as a percentage of GFP positive J-Lat cells (**C**). Representative dot plots obtained by flow cytometry represented in C (**D**). Blue corresponds to areas of lower cell density, yellow represents mid-range, and red indicates areas of high cell density. NT: non-treated, NI: non-infected. SN: supernatants. Data are expressed as mean ± SD obtained from 4 independent experiments. **** *p* < 0.0001 vs. cells NT, SN-LX-2(NI), J-Lat: LX-2 (NI).

**Figure 2 pathogens-13-00134-f002:**
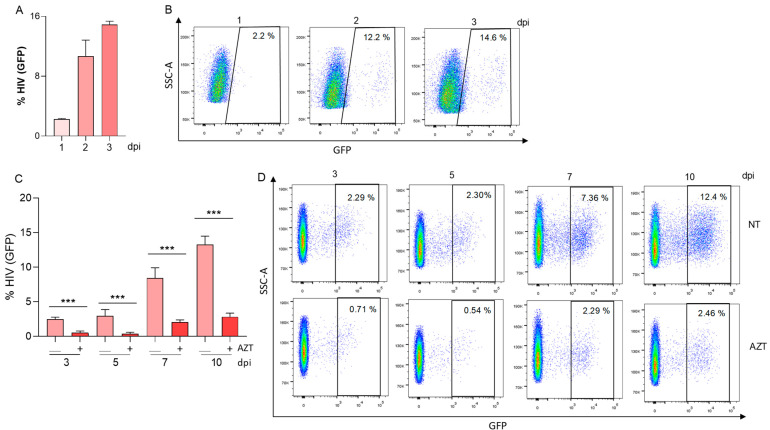
HIV replicate in LX-2 cells and produce infectious viral particles. Kinetics of HIV replication using (0.5 pg of p24 per cell) viral inoculum of VSV-G-HIV-GFP, measured as a percentage of GFP positive cells at 1, 2, and 3 DPI (days post-infection) (**A**). Representative dot plots obtained by flow cytometry represented in A (**B**). Jurkat cells were preincubated or not with AZT (azidothymidine) and exposed to culture supernatants from LX-2 cells infected with VSV-G-HIV-GFP (0.5 pg of p24 per cell) for 3 days. Viral replication was determined as the percentage of GFP positive cells measured at 3, 5, 7, and 10 dpi (days post-infection) (**C**). Representative dot plots obtained by flow cytometry represented in C (**D**). Blue corresponds to areas of lower cell density, yellow represents mid-range, and red indicates areas of high cell density. Data are expressed as mean ± SD obtained from 3 independent experiments. *** *p* < 0.001.

**Figure 3 pathogens-13-00134-f003:**
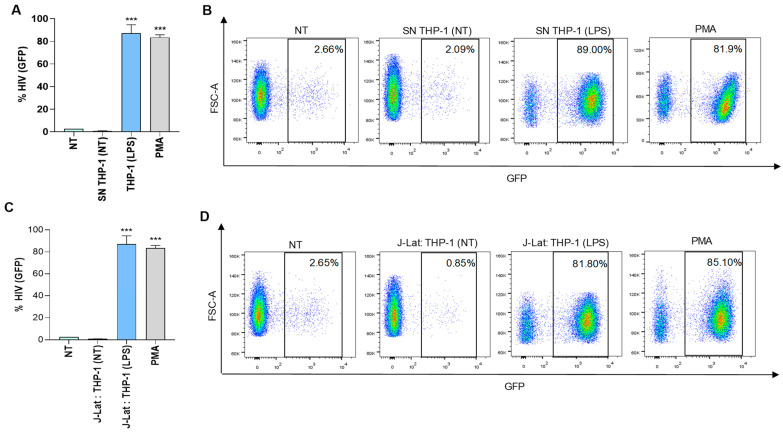
THP-1 cells stimulated with Lipopolysaccharide (LPS) were able to induce latency reversion in J-Lat cells. J-Lat cells were stimulated with culture supernatants from THP-1 cells stimulated with 100 ng/mL of Lipopolysaccharide (LPS) from *E. coli* at 1/2 dilution. Cells stimulated with 50 ng/mL of phorbol 12-myristate 13-acetate (PMA) were used as positive control. Culture supernatants from non-infected THP-1 cells were used as a control. At 72 h, latency reversion was quantified as a percentage of GFP positive J-Lat cells (**A**). Representative dot plots obtained by flow cytometry represented in A (**B**). LPS stimulated THP-1 were co-cultured with J-Lat cells at 1:1 proportion. Latency reversion was quantified as a percentage of GFP positive J-Lat cells (**C**). Representative dot plots obtained by flow cytometry represented in C (**D**). Blue corresponds to areas of lower cell density, yellow represents mid-range, and red indicates areas of high cell density. NT: non-treated, NI: non-infected. SN: supernatants. Data are expressed as mean ± SD obtained from 4 independent experiments. *** *p* < 0.001 vs. cells NT, SN THP-1(NT), J-Lat: THP-1 (NT).

**Figure 4 pathogens-13-00134-f004:**
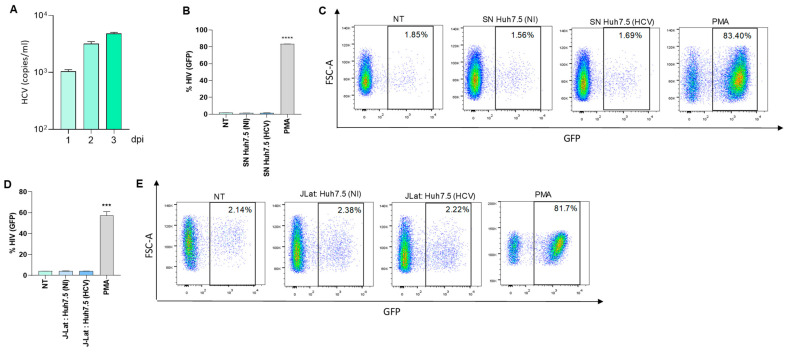
HCV-infected Huh7.5 cells were not able to reverse viral latency in J-Lat cells. Huh 7.5 cells were infected with HCV at a multiplicity of infection (MOI) = 1. Viral copies/mL were determined in culture supernatant by RT-qPCR at 1-, 2-, and 3-days post-infection (**A**). J-Lat cells were stimulated with culture supernatants from HCV-infected Huh7.5 cells, harvested at 24 h post-infection at 1/2 proportion. Cells stimulated with 50 ng/mL of phorbol 12-myristate 13-acetate (PMA) was used as a positive control. Culture supernatants from non-infected Huh7.5 cells were used as a control. At 72 h, latency reversion was quantified as a percentage of GFP positive J-Lat cells (**B**). Representative dot plots obtained by flow cytometry represented in B (**C**). HCV-infected Huh7.5 cells were co-cultured with J-Lat cells at 1:1 proportion. Latency reversion was quantified as a percentage of GFP positive J-Lat cells (**D**). Representative dot plots obtained by flow cytometry represented in D (**E**). Blue corresponds to areas of lower cell density, yellow represents mid-range, and red indicates areas of high cell density. NT: non-treated, NI: non-infected. SN: supernatants. Data are expressed as mean ± SD obtained from 4 independent experiments. *** *p* < 0.001, and **** *p* < 0.0001 vs. cells NT, SN-Huh7.5(NI); J-Lat: Huh 7.5 (NI).

**Figure 5 pathogens-13-00134-f005:**
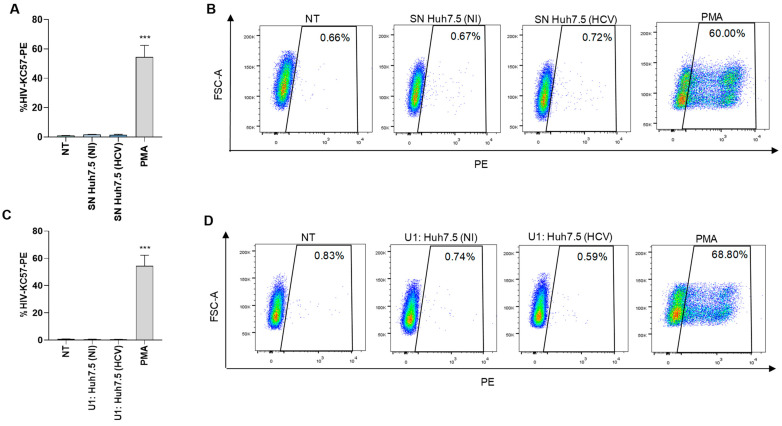
HCV-infected Huh7.5 cells were not able to reverse viral latency in U1 cells. U1 cells were stimulated with culture supernatants from HCV-infected Huh7.5 cells, harvested at 24 h post-infection at 1/2 proportion. Cells stimulated with 50 ng/mL of phorbol 12-myristate 13-acetate (PMA) were used as a positive control. Culture supernatants from non-infected Huh7.5 cells were used as a control. At 72 h, latency reversion was quantified by flow cytometry using a phycoerythrin (PE)-labeled KC57 monoclonal antibody against gag p24 and expressed as a percentage J-Lat positive cells (**A**). Representative dot plots obtained by flow cytometry represented in A (**B**). HCV-infected Huh7.5 cells were co-cultured with U1 cells at 1:1 proportion. Latency reversion was quantified as a percentage of PE-labeled U1 cells (**C**). Representative dot plots obtained by flow cytometry represented in C (**D**). Blue corresponds to areas of lower cell density, yellow represents mid-range, and red indicates areas of high cell density. NT: non-treated, NI: non-infected. SN: supernatants. Data are expressed as mean ± SD obtained from 4 independent experiments. *** *p* < 0.001 vs. cells NT, SN-Huh7.5 (NI), U1: Huh7.4 (NI).

## Data Availability

The raw data supporting the conclusions of this article will be made available by the authors, without undue reservation.

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
