# Peer review of "HIV-Infected Hepatic Stellate Cells or HCV-Infected Hepatocytes Are Unable to Promote Latency Reversal among HIV-Infected Mononuclear Cells"

_pathogens, 2024, doi:10.3390/pathogens13020134_

Round 1

Reviewer 1 Report

Comments and Suggestions for Authors

The authors establish a cellular experimental model to verify that cells latently infected by HIV do not reverse their latency in contact with cells or supernatants of cell cultures of HCV-infected hepatocytes. The study is very interesting and rules out that it is the hepatic parenchymal cell line. It is well built and the results are expressed accurately. I would only like that in the discussion you could analyze or hypothesize what happens in the liver of patients coinfected by HIV and HCV, once the involvement of the hepatocyte has been ruled out, since it is not the only cell that can harbor HCV and there are other factors that can participate, such as the bacterial LPS released in the bacterial translocation phenomena that are very frequent in this type of patients.

Author Response

Reviewer 1

Comments and Suggestions for Authors

The authors establish a cellular experimental model to verify that cells latently infected by HIV do not reverse their latency in contact with cells or supernatants of cell cultures of HCV-infected hepatocytes. The study is very interesting and rules out that it is the hepatic parenchymal cell line. It is well built, and the results are expressed accurately. I would only like that in the discussion you could analyze or hypothesize what happens in the liver of patients coinfected by HIV and HCV, once the involvement of the hepatocyte has been ruled out, since it is not the only cell that can harbor HCV and there are other factors that can participate, such as the bacterial LPS released in the bacterial translocation phenomena that are very frequent in this type of patients.

Author Response: We thank the reviewer for his/her keen observations. We have included the suggested issue in the discussion of the revised version. Please see the text highlighted in yellow in the discussion section lines 355-375.

Reviewer 2 Report

Comments and Suggestions for Authors

Lopez et al attempt to determine if co-culturing HIV or HCV infected liver cells with either CD4+ T cell or monocyte/macrophage latency models will reactivate HIV. They demonstrate that either direct co-culture or overlaying supernatants from infected cells does not induce latency reversal, whereas treating cells with the latency-reactivation agent PMA does. In general, this study needs to be further developed to support the conclusions being made. First, no experiments are performed to confirm that the LX2 liver cells are productively infected with either HIV or HCV. The assays used to titer these viruses are not indicative of infectious virus, as p24 ELISAs only measure HIV GagPol and not infectious virus. The same is true for only looking at HCV viral RNA. LX2 cells should have been characterized for production of viral proteins, viral mRNAs, or release of nascent particles to confirm that productive infections were occurring, especially given that all results presented are negative. Second, the “positive controls” used for these studies are apples-to-oranges comparisons. Treating latent cell models will strong reactivating chemicals does not indicate that the co-culture system is working. Likewise, you can’t treat THP1 cells with LPS, and then co-culture the THP1+LPS mixture with J-Lat cells and say that reactivation is from the activated THP-1 cells alone. Third, dose curves should have been performed with the infected LX2 cells to demonstrate that the lack of a response was not due to a single-dose effect, especially since no experiments were performed to determine the percent of LX2 cells productively infected. Hypothetically, if only 2% of these cells were infected, one would not anticipate seeing an impact on the latency models. However, if a dose curve were performed, perhaps latency reactivation could have been achieved. Likewise, its plausible that this LX2 model is not amenable to this experimental approach, and these observations are a cell-line specific artifact. Additional liver cell models should be tested in parallel to demonstrate that this isn’t a one-off artifact.

Major:

-The “positive controls” used in this study are really comparing apples to oranges. Yes, these LRAs can induce latent provirus in these cell models, but it is possible that this co-culture system is not working in general. While the authors use co-culture of THP-1 cells stimulated with LPS as an “apples-to-apples” comparison, there are no controls provided that demonstrate it is not residual/carryover LPS inducing activation of the Jurkat cells. Did the authors take cell media +LPS, in the absence of THP-1 cells, and add that to the J-Lat cells to demonstrate no induction? This study would benefit greatly from more direct controls to demonstrate the cell systems are working as the authors suggest.

- What other assays were performed to verify that LX-2 cells were productively infected with the pseudotyped virus? The results section does not give details and the method section simply states that “24-well plates were exposed to an HIV inoculum of 0.5 pg of p24 per cell.” Since the co-transfections were performed at a ratio of 10:1 for provirus to VSV-G, there is the potential that an over-abundance of non-infectious GagPol will be in the supernatant and give an inaccurate estimation of infectious virus solely using p24 ELISA. Along these lines, no validation is performed to determine HCV infection efficiency in these studies. Because everything in this study is based on these cells being infected, production of viral proteins, mRNAs, newly released virus etc. in the “infected” cells should be determined for both viruses.

- In addition to the point above, did the authors try a dose curve of infection with the LX-2 cells? Hypothetically, if that amount of p24 exposure only resulted in a few percent of the cells being infected, it is possible that a higher inoculum would be more revealing. Especially since all of the responses observed in this study are negative. Furthermore, are there any other hepatocyte lines that could be tested to rule out that this isn’t a cell line-specific effect?

- Why aren’t the details of the flow cytometry procedure listed in the materials and methods section? This becomes particularly important when the authors treat U1 cells with high concentrations of PMA, which would induce their differentiation and adherence to the tissue culture plate

Minor:

- The intro states that only 22 million HIV+ patients are on ART (that would be less than 60%), this number seems inaccurate. Can the authors reference a recent study where this is established? The study referenced is focused on patients 65 and older and doesn’t seem to support this statement. The WHO website indicates that as of 2022 almost 30 million patients are on ART.

- THP-1 and U1 cells are not considered macrophage-like under steady state conditions, as they are non-adherent and undergoing rapid cell division. These cell types need to be differentiated to induce their adherence/lose cell cycle progression to be considered macrophage-like. This should be clarified in the text

Comments on the Quality of English Language

English language is adequate

Author Response

Reviewer 2

Comments and Suggestions for Authors

Reviewer Comment 1: Lopez et al attempt to determine if co-culturing HIV or HCV infected liver cells with either CD4+ T cell or monocyte/macrophage latency models will reactivate HIV. They demonstrate that either direct co-culture or overlaying supernatants from infected cells does not induce latency reversal, whereas treating cells with the latency-reactivation agent PMA does. In general, this study needs to be further developed to support the conclusions being made. First, no experiments are performed to confirm that the LX2 liver cells are productively infected with either HIV or HCV.

The assays used to titer these viruses are not indicative of infectious virus, as p24 ELISAs only measure HIV GagPol and not infectious virus. The same is true for only looking at HCV viral RNA. LX2 cells should have been characterized for production of viral proteins, viral mRNAs, or release of nascent particles to confirm that productive infections were occurring, especially given that all results presented are negative.

Author Response: We appreciate this observation and concur with the reviewer's comment. The raised concern has been addressed in the revised version. The assessment of HIV infectivity and replication in LX-2 cells included measuring intracellular p24 expression using the KC57 monoclonal antibody labeled with phycoerythrin against p24 (PE-KC57 [FH190-1-1]) protein (Beckman Coulter, United States) through flow cytometry at 24, 48, and 72 hours. Additionally, infective particles were released by LX-2 cells, as demonstrated when culture supernatants harvested 72 hours after LX-2 infection were able to infect Jurkat cells. Moreover, pretreatment of Jurkat cells with AZT significantly reduced viral replication, providing further evidence for the infectivity of the viral progeny released from infected LX-2 cells. Please refer to the highlighted text in yellow (lines 255-265 and Figure 2).

To assess HCV infectivity and replication in Huh7.5 cells, culture supernatants obtained at 24, 48, and 72 hours after virus inoculation were analyzed. Replication kinetics were evaluated using an in-house RT-qPCR and confirmed by a commercial assay (Quantitative Real-Time PCR-based Cobas® HCV Test). Please see the text highlighted in yellow in the revised version of the manuscript, lines 299 and 300 and figure 4A.

Reviewer comment 2: Second, the “positive controls” used for these studies are apples-to-oranges comparisons. Treating latent cell models will strong reactivating chemicals does not indicate that the co-culture system is working. Likewise, you can’t treat THP1 cells with LPS, and then co-culture the THP1+LPS mixture with J-Lat cells and say that reactivation is from the activated THP-1 cells alone.

Author Response: We thank this reviewer observation. Co-culture experiments were not performed in the presence of a mixture of THP-1 cells plus LPS and J-Lat. After 4 hours post LPS stimulation THP-1 cells were washed to eliminate the remain extracellular LPS. This point was clarified by rewrite the paragraph as follow:

THP-1 cells were stimulated for 4 hours with E. coli lipopolysaccharide (LPS), and then pelleted by centrifugation at 180g for 15 min and washed twice in 10 ml of RPMI. Then these cells were co-cultured with J-Lat cells. Alternatively, culture supernatants from THP-1 cells stimulated with LPS for 4 hours, and then pelleted by centrifugation at 180g for 15 min and washed twice in 10 ml of RPMI and then cultured for an additional 18 hours in fresh culture medium and used to stimulate J-Lat cells. Therefore, a direct role of E. coli LPS could be rule out. Please see lines highlighted in yellow (119-125).

Reviewer comment 3: Third, dose curves should have been performed with the infected LX2 cells to demonstrate that the lack of a response was not due to a single-dose effect, especially since no experiments were performed to determine the percent of LX2 cells productively infected. Hypothetically, if only 2% of these cells were infected, one would not anticipate seeing an impact on the latency models. However, if a dose curve were performed, perhaps latency reactivation could have been achieved. Likewise, its plausible that this LX2 model is not amenable to this experimental approach, and these observations are a cell-line specific artifact. Additional liver cell models should be tested in parallel to demonstrate that this isn’t a one-off artifact.

Author Response: As indicated in response to reviewer comment 1, we assessed the percentage of LX-2 infected cells, and the infection efficiency reached 14.95± 0.93% at 72 hours post-infection.

The LX-2 human hepatic stellate cell line retains key features of HSCs (PMID: 15591520). We agree with the reviewer that incorporating other cell models could enhance this manuscript. However, since we demonstrated that VSV-G-HIV was capable to replicate in LX-2 cells this experiment is not mandatory.

Additionally, previous studies indicated lower replication of HIV in hepatocytes, despite the presence of integrated HIV-DNA.

(Ling Kong 1, Walter Cardona Maya, Maria E Moreno-Fernandez, Gang Ma, Mohamed T Shata, Kenneth E Sherman, Claire Chougnet, Jason T Blackard. Low-level HIV infection of hepatocytes. Virol J. 2012 Aug 9:9:157. doi: 10.1186/1743-422X-9-157.)

Major:

Reviewer comment 4: -The “positive controls” used in this study are really comparing apples to oranges. Yes, these LRAs can induce latent provirus in these cell models, but it is possible that this co-culture system is not working in general. While the authors use co-culture of THP-1 cells stimulated with LPS as an “apples-to-apples” comparison, there are no controls provided that demonstrate it is not residual/carryover LPS inducing activation of the Jurkat cells. Did the authors take cell media +LPS, in the absence of THP-1 cells, and add that to the J-Lat cells to demonstrate no induction? This study would benefit greatly from more direct controls to demonstrate the cell systems are working as the authors suggest.

Author Response: While we partially agree with the reviewer on this point, it is important to note that, as indicated in the response to Reviewer comment 2, cells were washed after 4 hours of LPS stimulation. We acknowledge the possibility of residual LPS in the supernatants used to stimulate J-LAT cells. However, considering that comparable levels of latency reversion were observed in J-LAT cells stimulated with culture supernatants from cells stimulated with LPS and in the coculture model, the likelihood of a direct effect of LPS can be effectively ruled out.

Reviewer comment 5: - What other assays were performed to verify that LX-2 cells were productively infected with the pseudotyped virus? The results section does not give details and the method section simply states that “24-well plates were exposed to an HIV inoculum of 0.5 pg of p24 per cell.” Since the co-transfections were performed at a ratio of 10:1 for provirus to VSV-G, there is the potential that an over-abundance of non-infectious GagPol will be in the supernatant and give an inaccurate estimation of infectious virus solely using p24 ELISA. Along these lines, no validation is performed to determine HCV infection efficiency in these studies. Because everything in this study is based on these cells being infected, production of viral proteins, mRNAs, newly released virus etc. in the “infected” cells should be determined for both viruses.

Author Response: We thank this reviewer observation. Infectivity and replication of each virus was evaluated as was indicated in the response to Reviewer comment 1

Reviewer comment 6: - In addition to the point above, did the authors try a dose curve of infection with the LX-2 cells? Hypothetically, if that amount of p24 exposure only resulted in a few percent of the cells being infected, it is possible that a higher inoculum would be more revealing. Especially since all of the responses observed in this study are negative. Furthermore, are there any other hepatocyte lines that could be tested to rule out that this isn’t a cell line-specific effect?

Author Response: Please see response to comment 1 and 3.

Reviewer comment 7: - Why aren’t the details of the flow cytometry procedure listed in the materials and methods section? This becomes particularly important when the authors treat U1 cells with high concentrations of PMA, which would induce their differentiation and adherence to the tissue culture plate.

Author Response: We agree with the reviewer on this point. The description of the flow cytometry procedure has been rewritten to clarify this aspect. Please refer to Section 2.4 in the Materials and Methods of the revised version of the manuscript, lines 204-214.

Minor:

Reviewer comment 8: - The intro states that only 22 million HIV+ patients are on ART (that would be less than 60%), this number seems inaccurate. Can the authors reference a recent study where this is established? The study referenced is focused on patients 65 and older and doesn’t seem to support this statement. The WHO website indicates that as of 2022 almost 30 million patients are on ART.

Author Response: We agree with the reviewer on this point. The sentence has been corrected in the revised version of the manuscript. Please see the text highlighted in yellow (lines: 45-50)

Reviewer comment 9: - THP-1 and U1 cells are not considered macrophage-like under steady state conditions, as they are non-adherent and undergoing rapid cell division. These cell types need to be differentiated to induce their adherence/lose cell cycle progression to be considered macrophage-like. This should be clarified in the text.

Author Response: U1 is a cell line displaying inducible levels of HIV expression obtained from the surviving population of U937 cells acutely infected with the X4 HIV-1LAI/IIB strain. U1 cell line become prototypic models to delineate the regulatory effects of host determinants.

PMA stimuli reverse the HIV latency in U1 cells. Therefore, these cells were used without PMA pretreatment.

(Folks, T. M., Justement, J., Kinter, A., Schnittman, S., Orenstein, J., Poli,G., Fauci, A. S. (1988) Characterization of a promonocyte clone chronically infected with HIV and inducible by 13-phorbol-12-myristate acetate. J. Immunol. 140, 1117–1122.

Folks, T. M., Justement, J., Kinter, A., Dinarello, C. A., Fauci, A. S.(1987) Cytokine-induced expression of HIV-1 in a chronically infected promonocyte cell line. Science 238, 800–802.

Edana Cassol, Massimo Alfano, Priscilla Biswas, Guido Poli. Monocyte-derived macrophages and myeloid cell lines as targets of HIV-1 replication and persistence J Leukoc Biol. 2006 Nov;80(5):1018-30. doi: 10.1189/jlb.0306150.)

When stimulated with PMA, THP-1 cells mimic MDM by becoming adherent to glass or plastic, exhibiting a macrophage-like morphology, and expressing macrophage differentiation markers.

However, these cells respond adequately to LPS, like monocytes, producing proinflammatory cytokines and becoming partially and transiently adherent.

THP-1 transiently adhered in response to LPS induction. Maximum cell attachment occurred at 1 hour after LPS stimulation, and the number of attached cells returned to baseline by 4 hours, consistent with previous descriptions.

(Nicole S Kounalakis, Siobhan A Corbett. Lipopolysaccharide transiently activates THP-1 cell adhesion. J Surg Res. 2006 Sep;135(1):137-43. doi: 10.1016/j.jss.2005.12.018.)

Reviewer 3 Report

Comments and Suggestions for Authors

The authors conducted a study of cellular mechanisms of HIV and HCV co-infection. The study is both well written and informative. I primarily have recommendations to improve the clarity of the manuscript.

  1. Overall, the intro should make a stronger case for the importance of understanding HIV/HCV interaction. Some additional background on the cumulative effects of HIV/HCV coinfection would be helpful to address this.
  2. In the last sentence of the first paragraph in the intro, I recommend specifying that the improvements in life expectancy are relative to those not taking HAART. It’s mostly intuitive as is, but this does add a bit of clarity.
  3. In the statistical analysis section, rather than say that analyses were performed where applicable, it think it’s fine to specify what statistical comparisons are being made (for example, reiterating what the groups being compared are).
  4. Was a standard Tukey’s test used for pairwise mean comparisons, or was an adjustment for ordinal data included? Note that the standard Tukey’s comparison is fine if your means weren’t overly skewed, but if they were you may want to consider a multi group ordinal test (like a Kruskal-Wallis test) with a pairwise post-doc analysis.
  5. A citation for your statistical software should be added.
  6. Implications for research practice should be more elaborated on.
  7. Additional future recommendations for related research would be helpful. 

Author Response

Reviewer 3

Comments and Suggestions for Authors

The authors conducted a study of cellular mechanisms of HIV and HCV co-infection. The study is both well written and informative. I primarily have recommendations to improve the clarity of the manuscript.

Reviewer comment: Overall, the intro should make a stronger case for the importance of understanding HIV/HCV interaction. Some additional background on the cumulative effects of HIV/HCV coinfection would be helpful to address this.

In the last sentence of the first paragraph in the intro, I recommend specifying that the improvements in life expectancy are relative to those not taking HAART. It’s mostly intuitive as is, but this does add a bit of clarity.

Author Response: We appreciate the suggestions made by the reviewer. Additional background information related to HIV/HCV coinfection and the improvements in life expectancy associated with HAART has been included. Please review the highlighted paragraphs in yellow (Lines 48-50 and 77-81).

Reviewer comment: In the statistical analysis section, rather than say that analyses were performed where applicable, it think it’s fine to specify what statistical comparisons are being made (for example, reiterating what the groups being compared are).

Was a standard Tukey’s test used for pairwise mean comparisons, or was an adjustment for ordinal data included? Note that the standard Tukey’s comparison is fine if your means weren’t overly skewed, but if they were you may want to consider a multi group ordinal test (like a Kruskal-Wallis test) with a pairwise post-doc analysis.

Author Response: We accept the reviewer's feedback. We have provided details on the specific comparisons conducted in the statistical analysis. Part of the statistical analysis description was incomplete and has been corrected in the revised version of the manuscript. ANOVA is initially performed to confirm significant differences between groups before applying the Tukey test. Please refer to the highlighted text in yellow for further clarification (Lines 216-226).

Reviewer comment: A citation for your statistical software should be added.

Author Response: Thank you. The complete citation has been included. Please see lines 225-226.

Reviewer comment: Implications for research practice should be more elaborated on. Additional future recommendations for related research would be helpful.

Author Response: The concerns were addressed, and the discussion of the manuscript was substantially improved. Please refer to the text highlighted in yellow for these enhancements (Lines 433-446).

Round 2

Reviewer 2 Report

Comments and Suggestions for Authors

The authors have addressed all of my previous comments and concerns by adding additional data and providing more details in the methods. I have no further comments or concerns